# Using the Fourier Methods for Cycle Counting of Bimodal Stress Histories with Variable in Time Amplitudes of Components

**DOI:** 10.3390/ma16010254

**Published:** 2022-12-27

**Authors:** Marek S. Kozień

**Affiliations:** Department of Applied Mechanics and Biomechanics, Faculty of Mechanical Engineering, Cracow University of Technology, 31-155 Krakow, Poland; marek.kozien@pk.edu.pl

**Keywords:** spectral method, multiaxial fatigue, mean stress effect, bimodal stresses, fatigue life

## Abstract

The application of the Fourier methods to perform cycle identification and description for different cases of bimodal stress histories are presented and discussed in the paper. The direct spectral method and the modified direct spectral method, previously co-proposed by the author, together with the aspects of the use of Fourier methods discussed in this article, offer a unique alternative to the methods known in the literature for identifying and counting various types of bimodal stress variations in time (with constant or variable amplitudes; completely reversed, reversal, or pulsed type). The application of the Fast Fourier Transform (FFT) and the Short-Time Fourier Transform (STFT) is discussed. The method is useful, especially in cases when it is known that due to the form of work of the mechanical system the two existing components of vibrations with various frequencies can be identified, of which the one with higher frequency has a vibration amplitude lower than a component with a lower frequency and, above all, a variable in time.

## 1. Introduction

The vibration of engineering structures often produces a few-modal type of stress history. The most common is the dominant one frequency history (one-modal case). All methods of fatigue cycle-counting are referred to this case. The second type of considered cases is the bimodal one. This type can be observed, for example, in a ship’s hull when it is forced to vibrate by the propulsion system with one dominant frequency and by the impact of sea waves [1], or in the suspension of the car [2]. In previous research, the case of the tri-modal type has been separated in fatigue analysis [3]. An example of the application is a new tri-modal model under stochastic loads by C. Han, X. Qu, S. Ding, and Y. Ma [4].

This article relates to the case of bimodal vibrations generating a bimodal history of stress variability in structure. The literature presents the results of scientific research on the issues of fatigue that analyze the bimodal process. S. Sakai and H. Okamura [5] analyzed a case with two dominant frequencies, clearly spaced from each other. It was a generalization of previous work on the problem in which a narrow frequency band was associated with one frequency [6,7]. The disadvantage of this proposed approach was the lack of consideration of the increase in the stress amplitude of the primary cycle with low frequency due to the occurrence of a component with a higher frequency. Therefore, T.-T. Fu and D. Cebon proposed their own concept of the counting of stress cycles that was based on the idea of taking into account the superposition of the amplitudes of harmonic components with low and high frequencies. Based on the above-mentioned methods, D. Benasciutti and R. Tovo [8] proposed the so-called modified Fu-Cebon method. The other attempt dedicated to the case of the superposition of deterministic-excited vibrations with impulse-excited ones whilst considering the damping effect was proposed by G. Jiao and T. Moan [9]. C. Han, Y. Ma, X. Qu, and M. Yang [10] were focused on the above-mentioned three methods: Jiao-Moan, Fu-Cebo,n and modified Fu-Cebon. It has been pointed out that these bimodal methods are based on the knowledge of the frequency characteristics of stress and not the variation of stress in the time domain. Moreover, in the article [10] an analytical solution for predicting the vibration-fatigue-life in bimodal spectra is developed based on random attempts.

M.S. Kozień and D. Smolarski formulated their own method for the cycle-counting of bimodal stress of completely reversed type, based on their frequency characteristics. The method was formulated at the beginning for uniaxial stress [11,12] and later for multiaxial stress [13]. In the article [14], P.J. Romanowicz, D. Smolarski, and M.S. Kozień formulated and applied the modified direct spectral method. The idea was to apply the correction factor taking into account the influence of the mean compressive stresses. The method was presented with the example of a thrust roller bearing, using the results of numerical simulations of stress variation in time and based on analysis of the time domain. It was concluded that the proposed method (in contrast to multiaxial high-cycle fatigue criteria) included both kinds of stress waveforms in the fatigue analyses and could be applied for the identification of stress components and determination of fatigue life. It should be noted that, as far as the analysis presented in this article is concerned, the aim of which is to show the application of Fourier methods for identifying and describing cycles, the approaches to the direct spectral method [11,12,13] and the modified direct spectral method [14] presented in these articles are no different.

In engineering practice, the form of stress variability (stress tensor) is usually known. It can be ascertained directly from measurements carried out on the real object (for the structure or sample element), from the analytical solution of the problem, or as the result of computer simulations carried out based on the structure calculation model used. To identify the amplitude-frequency characteristics, it is common to use Fourier analysis methods of signal, in particular Fourier Series (FS) and Fast Fourier Transform (FFT) for stationary signals and Short Time Fourier Transform (STFT) for non-stationary ones. This article discusses the problem of using the FFT and STFT functions to properly identify the amplitude values of bimodal components, considering on the one hand the stress functions with constant or variable amplitudes, and on the other hand the components of the completely reversed type, reversal, as well as the pulsed type. While these types of courses are often found in practice, in particular the analysis of the variable amplitude course is not discussed in literature. In these cases, it is suggested to use one of the methods of counting irregular stress cycles, most often the rain-flow method. In the case of the bimodal cases, these methods seem to be unnatural. Another possible approach is the application of dedicated spectral methods. However, also in this case, the variability of the amplitude of stress in time is difficult to identify, especially considering that in the fatigue analysis, the averaged amplitude of stress variability is not enough to analyze. The key role is played by a maximal range of variability, and the identification of such cycles should be focused upon above all. The direct spectral method and the modified direct spectral method, previously co-proposed by the author, together with the aspects of the use of Fourier methods discussed in this article, provides a unique alternative to the methods known in the literature for identifying and counting various types of bimodal stress variations in time (with constant or variable amplitudes; completely reversed, reversal, or pulsed type).

In the fatigue life analysis of the structure, the identified stress cycles of various types, usually characterized by their number, average stress value, and stress amplitude value, are data for further calculations. Different methods of cumulative damage are used for this purpose. Commonly used in engineering applications is the Palmgren-Miner linear cumulative damage rule [15,16]. The method is simple in its principle but has its limitations in application. Therefore, different non-linear approaches are currently used, e.g., the double linear damage rule; Marco and Starkey; Subramanyan, Hashin, and Rotem; Corten and Dolon; Freudenthal-Heller; Serensen; Subramanayan; Bui-Quoc [17]; and Liou [7]. This aspect of fatigue analysis is not the purpose of this article.

The research question posed in this article is as follows: to what extent is it possible and effective to use Fourier’s methods of function analysis, in combination with the direct and modified direct spectral method, to count stress cycles with variable amplitudes in time, of the bimodal type, whose courses are irregular in time.

The paper is divided into five sections. A state of the art introduction to the bimodal stress fatigue problem is given in Section 1. The theoretical background and interpretation of the application of the Fourier methods (FS, FFT, STFT) is discussed in Section 2. The theoretical background of direct spectral methods is described in Section 3. Cycle identification using FFT and STFT methods for three specially selected examples of stress variation over time, involving a complex combination of types of variation (reversal, pulsed) and variation of amplitude of the higher frequency component is thoroughly discussed in Section 4. The two examples of stress variation are specially generated for analysis, and the third refers to the stresses generated in a ball bearing. The results obtained using the rainflow cycle counting method, commonly used for irregular stress histories, and the direct spectral method, using FFT or STFT for identification process, were compared. Finally, the paper is concluded in Section 5.

## 2. Identification of Pulsating Waveform Based on Fourier Analysis

The commonly used method for identifying the frequency characteristic of the function is the so-called Fourier analysis. For stationary functions (signals in Theory of signals), it contains Fourier Series (FS) and Fourier Transform (FT). In computer realization commonly used in algorithm of FT is the Fast Fourier Transform (FFT). For non-stationary ones, the Short Time Fourier Transform (STFT) is used. Let us define function *x(t)* in the range *t*∈[0,*T*], assuming the fulfillment of the Dirichlet conditions. It can be shown in the form of infinite series (1), called as Fourier Series (FS) representation of *x(t)*, where: $a_{0}=\frac{1}{T}\overset{T}{\underset{0}{\int }}x\left(t\right)dt$, $a_{n}=\frac{2}{T}\overset{T}{\underset{0}{\int }}x\left(t\right)\sin\left(n\omega t\right)dt,$ $b_{n}=\frac{2}{T}\overset{T}{\underset{0}{\int }}x\left(t\right)\cos\left(n\omega t\right)dt,$ $\omega =\frac{2\pi }{T}$, n=1,…,+∞.
(1)$$x\left(t\right)=a_{0}+\overset{+\infty }{\underset{n=0}{\sum }}\left[a_{n}sin\left(n\omega t\right)+b_{n}cos\left(n\omega t\right)\right]$$

Each function *x(t)*, for which the integral of its absolute value over the real space ***R*** is finite, can be transformed into the frequency value using the Fourier Transform (2), where *I* is the imaginary unit. The FT Fourier Transform (FT) is defined in the complex space. Usually, the Fast Fourier Transform (FFT) algorithm is used in this case.
(2)$$FT\left[x\right]\left(\omega \right)=\frac{1}{\sqrt{2\pi }}\overset{+\infty }{\underset{-\infty }{\int }}x\left(t\right)e^{-i\omega t}dt$$

When the function is non-stationary, the generalization of the FT (2) is attained by inserting the specially defined window function *g(t)* centered for time τ running along time domain (independent time variable), defining the STFT function of *x(t)* (3).
(3)$$STFT\left[x\right]\left(\omega ,t\right)=\frac{1}{\sqrt{2\pi }}\overset{+\infty }{\underset{-\infty }{\int }}x\left(\tau \right)g\left(\tau -t\right)e^{-i\omega \left(\tau -t\right)}d\tau$$

The FS and especially FT in FFT realization is commonly used in identifying the frequency form of the signals in the dynamic analysis of structures for stationary functions and STFT for non-stationary ones. In particular, they can be applied to the identification or verification of the bimodal form of the signal.

In fatigue analysis, the most commonly analyzed stress variation in time domain has the form of a completely reversed form. Sometimes, however, the pulsating form is recognized. It should be noted that the spectral forms of the two types of realization have completely different forms in Fourier analyses. Therefore, it is very important to properly interpret the Fourier spectras of stress signals, because the direct spectral method is based on reconstruction of the two harmonic components based on the spectrum of the stress signal.

Let us consider the two test signals—the harmonic sine one modeling the completely reversed stress (4) and the signal in the form of absolute value of harmonic sine modelling the pulsating (zero-to-tension or zero-to compression) stress (5).
(4)x(t)=Asin(ωt)
(5)x(t)=A|sin(ωt)|

The time-history of the harmonic function (4) and module of its FFT function is shown in Figure 1. The second pair of graphs depicting two cases of analysis of the signal (5) is shown in Figure 2 and Figure 3. In Figure 3 the signal in the form (5) shifted vertically after the procedure subtract mean was applied. The subtract mean procedure consisted of shifting the graph of the function (function course) in the vertical direction (change of function value) in such a way that, finally, the average value of the function in the analyzed interval was equal to zero.

The following remarks relate to the corresponding forms of the FS representations:For the signal (4) the values of coefficients are:

$a_{0}=0$,

$a_{1}=1,\;a_{n}=0,n=2,\ldots ,+\infty$,

$b_{n}=0,n=1,\ldots ,+\infty$.

For the signal (5) the values of coefficients are:

$a_{0}=0.662$,

$a_{n}=0,n=1,\ldots ,+\infty$,



$b_{1}=0.882,\;b_{2}=0.212,\;b_{3}=0.08,\ldots$



The most important differences in the FFT spectras of signals (4) and (5) are:The harmonic function (4) has only the single strip for the given frequency of the signal for the function;The absolute value of harmonic function (5) has a series of strips with decreasing amplitudes with non-zero value for zero-frequency and multiplied (especially doubled) basic frequency.

## 3. Formulation of the Direct Spectral Method of Cycle Counting

The direct spectral method and the modified direct spectral method are dedicated to cases where the variability of bimodal stresses is the source of structure vibrations. The methods are based on the following assumptions:The primary cycle is of a low-frequency character with a frequency ω_1_ and a larger constant amplitude *A_1_* than the higher-frequency component with a frequency ω_2_ and the time-varying amplitude *A_2_(t)*.In general, the frequency ω_2_ is much greater than the frequency ω_1_ and the amplitude A_1_ is much greater than the maximum value of the amplitude *A_2_(t)*.The analysis is performed for the stress block period *T_B_*, calculated based on the values of *T_1_* and *T_2_*, where *T_B_* is the smallest total multiple of the *T_1_* period for which the ratio *T_B_/T_1_* is the approximate integer.Generally, the variation of stress for low-frequency cycles has harmonic shape; if realistic forms is other (identified in computer simulation or measurement) the dedicated analytical function can be used for an approximation of this shape.High-frequency components varying with frequency ω_2_ generates the so-called secondary cycles.The difference between the direct spectral method and the modified direct spectral method is the formula (the applied theory) that is used to consider the average stress value and determine finally the equivalent completely reversed stress. The direct spectral method was applied in the analyses discussed in [11,12,13] and the modified direct spectral method is presented in [14].The difference in the use of direct spectral methods [11,12,13,14] and the spectral methods known in the literature [1,2,3,4,5,6,7,8,9,10], especially [7], lies in the description. The formulation of the direct spectral methods is deterministic, and the spectral methods are random. It seems, therefore, that direct methods are more natural in the analysis of bimodal waveforms. However, the key here is to identify the amplitudes and average stress values of the main cycle and secondary cycles. Fourier methods (FFT, STFT) can be used for this purpose.The algorithm of direct spectral methods was successively developed and presented in articles [11,12,13,14]. In particular, article [14] contains a detailed algorithm for the last modified direst spectral method.

## 4. Results and Discussion of Application of Fourier Methods to Identify Stress Cycles for Several Dedicated Examples

### 4.1. General Remarks

To test the applicability and limitations of the Fourier analysis for quantitative identification of the parameters of the identified stress cycles, an analysis was performed for three cases of bimodal stress cycles (time signals). These three cases are as follows:Analytical (generated based on the analytical formula). Superposition of the absolute value of sine with constant amplitude for frequency *ω_1_* (pulsating form, zero-to-tension) and sine with constant amplitude for frequency *ω_2_* (reversal stress).Analytical (generated based on the analytical formula). Superposition of the absolute value of sine with constant amplitude for frequency *ω_1_* (pulsating form, zero-to-tension) and sine with amplitude linearly varying in time for frequency ω_2_. For the first half of period T_1_ the amplitude grew to maximum value *(A_2_)_max_* = *A_2_(T_1_/*2*)*, and for the second half of period *T_1_*_,_ it fell to zero.Numerical simulation based on solution of the realistic problem of stresses existing in rolling bearings due to repeated contact between rollers and rings. The signal was one of the components of stress tensor analysed and discussed by P.J. Romanowicz, D. Smolarski, and M.S. Kozień [14]. Due to the nature of the contact, the component with frequency ω_1_ had a zero-to-tension form, but the variation in time did not have a sine form, and had to be approximated in a suitable form. Moreover, the component with frequency ω_2_ *ω_2_* had a sine form with various in time amplitude that grew from zero to maximum value in the middle of the period T_1_ *T_1_*, before falling to zero. The variability of amplitude A_2_ in time depended on the rolling process of the ball in contact with the bearing raceway and was not defined in analytical form.

For all cases, the values of the average stress and stress amplitude were compared with those obtained using the rainflow method [18], commonly used in fatigue analyses. The VibrationData Toolbox ver. 13.3 by Tom Irvine in Partnership with ENDAQ was used for the analysis, in which the procedure was carried out in accordance with the standard ASTM E 1049-85. The values of the stress amplitude and mean stress were determined using the Fourier methods for the identified stress cycles and then the direct spectral method (called FOURIER in Tables); those obtained using the rainflow method (called RAINFLOW in Tables) were then compared. For the rainflow method, the following values were given: the average (Ave.) and maximum (Max.) value of the stress amplitude, as well as the minimum (Min.) and maximum (Max.) values of the mean stress value.

### 4.2. Analytical Simulation—Stationary Subcycles

The first example was devoted to the stress variation defined in the analytical form (8). The primary cycle had the pulsating form of absolute value of sine with amplitude of 800 MPa and frequency of 0.5 Hz (period equal to *T_1_* = 2 s). Due to the action of the absolute value type function, the final frequency of the signal was equal to 1 Hz with period equal to *T_B_* = 1 s (see Figure 4—left). The secondary cycles were reversal stress with amplitude of 200 MPa and frequency of 12.5 Hz. The visualization of the signal σ(t), named as *sigma*(t) on the plot, is shown in Figure 4 (left). The graphical form of FFT function of the function (6), named as *SIGMA*(f) on the plot, is shown in Figure 4 (right).
(6)σ(t)=800|sin(πt)|+200sin(25πt)

Identified cycles based on the applied spectral method are summarized in Table 1 (RAINFLOW denotes results obtained using the rainflow method and FOURIER those obtained by application of the Fourier transform and the direct spectral method). It should be emphasized that in the analyzed example, the stress block period was exactly equal to *T_B_* = 2 s. Comparison of different identified parameters of mean values and amplitudes of cycles, based on the rainflow method and application of the direct spectral method for stress variation defined by (6) is summarized in Table 1.

Due to constant amplitudes of the two components of the signal *σ*(t) (6), the FFT frequency analysis identified the correct amplitude and frequency values. The frequency of the first component was correctly identified as 1 Hz. The read-up value of 339.6 MPa was multiplied by a scale factor with a value equal to approximately 1/0.422, which gave a result of an amplitude of 804.6 MPa, which is very close to the value resulting from the analytical form of a function of 800 MPa. The value of a scale factor for a close 0.422 form is explained in the discussion presented in Section 2 referring to the FFT function for the sinusoidal course and the module with the sine function and can be identified on Figure 2 (right). The frequency of the second component was correctly identified as 12.5 Hz. However, the amplitude value read was 199 MPa and was very close to the value resulting from the analytical form of a function of 200 MPa. In this case, the value of the amplitude resulting from the FFT analysis was not multiplied by a scaling factor, because its source was a sine wave. This interpretation also resulted from the discussion presented in Section 2 referring to the FFT function for the sine wave and can be identified in Figure 1 (right). From the analysis of the case under consideration, it can be seen that for the correct identification of amplitude-frequency, it is necessary to reference quality to the course of the function *σ*(t) in the time domain.

Both methods identify approximately the same number of cycles (rainflow—26.5, Fourier—27). The Fourier method gave slightly higher stress amplitudes for the secondary cycles. On the other hand, the rainflow method additionally identified a stress half-cycle with a relatively high amplitude, which could not be identified on the time course. This approach is on the safe side and is conservative. On the other hand, the stress amplitudes identified for the two twin primary cycles had slightly higher values in the rainflow method. These results were not justified considering the time course of the function.

The identified values seem to be at a good level of agreement. The identification accuracy can also be increased by using the STFT method.

### 4.3. Analytical Simulation—Non-Stationary Subcycles

The second case was devoted to the stress variation whose envelope was, as in the previous example, the absolute value of sine with amplitude of 800 MPa and frequency of 0.5 Hz. The secondary cycles were reverse stress of sinus type with linearly variable amplitude starting with zero values for zero time, a maximal value of 200 MPa for mid-period time, zero value at the end of the period, and frequency of 12.5 Hz. Analytical form of the signal was defined by formula (7) where the weighting function was given by (8). The visualization of the function *σ*(t), named as *sigma*(t) is shown in Figure 5 (left).
(7)σ(t)=800|sin(πt)|+200w(t)sin(25πt)
(8)$$w\left(t\right)=w\left(t+T_{1}\right)=\{\begin{array}{cc} 2t & t\in [0,T_{1}/2)\\ 2\left(-t+1\right) & t\in [T_{1}/2,T_{1}] \end{array}$$

Because the second component of the signal with frequency ω_2_ had amplitude varying in time, it was reflected in the transformation of the FFT function (see Figure 5—right). The identified frequency value (1 Hz) and amplitude 399 MPa (corresponding in reality to 799 MPa) was the same as in the case under consideration in Section 4.2. This component had constant amplitude. On the other hand, the identified frequency for the second component was located around 12.5 Hz, but the identified value of the amplitude was about 100 MPa and so was lower than the maximum value of 200 MPa. This is because in the entire analyzed period, the amplitude of this ingredient changed over time, and the obtained appointment in FFT analysis was a certain averaging in the entire period of the course analysis. This means that, for this case, the FFT function could not be used to identify the amplitude of the variability of stress amplitudes, especially given the fact that in fatigue analysis the identification of the range of the highest variable stress was of key importance. In this case, with the supported identification of the amplitude of the second component as a function of time, it was necessary to use the STFT function. The graphical form of STFT function is shown in Figure 6.

Identified cycles based on the applied spectral method are summarized in Table 2 (RAINFLOW denotes results obtained using the rainflow method and FOURIER those obtained by application of the Fourier transform and the direct spectral method). The stress block period was exactly equal to *T_B_* = 2 s, as in the case analysed in Section 4.2. Comparison of different identified parameters of mean values and amplitudes of cycles, based on the rainflow method and application of the direct spectral method for stress variation defined by (7), are summarized in Table 2. The number of identified cycles was slightly higher for the Fourier method (by four cycles), but the amplitude values of these cycles were small and insignificant in fatigue analyses for a sample made of aluminum or steel. The identified stress amplitude values for primary cycles were very close in value. The stress amplitude values were generally at the same level for identified cycles. For cycles with smaller amplitudes, the Fourier method provided more conservative results.

The identified values seem to be at a good level of agreement. Accuracy of the identification can also be increased by using the STFT method with different parameters. This is possible by using a shorter time discretization of the time series of the stress course, which is built for the needs of discrete STFT analysis.

### 4.4. Stress Component of Rolling Bearing

The third example to show the possibility of using Fourier methods to identify irregular (with varying amplitudes in time) stress cycles refers to the stresses generated in the bearing balls during their operation. The issue of estimation of emerging stresses was discussed in articles [19,20]. However, in article [14] the possibility of using the modified spectral method for this purpose was shown, and its advantages over other commonly used identification methods were also shown, in particular considering compressive stresses. The analyses concerning the application of the direct method presented in article [14] referred to the comparison of methods and were based on the analysis of stress courses over time, obtained based on computer simulations. In the presented analysis concerning the use of Fourier methods in the implementation of the direct spectral method, the time course of the most difficult in the analysis case of the variability of the stress tensor component *σ*_z_ in the bearing balls was used. Figure 7 (left) shows variations in time of the analyzed component of stress tensor σ_z_ named as *sigma*(t); Figure 7 (right) shows its FFT representation, named as *SIGMA*(f). Due to the non-stationary character of the function for the proper data in frequency domain, which are varying in time, the STFT analysis must be applied. The results of STFT analysis are shown in Figure 8.

For the analyzed case of the stress component σ_z_ caused by the pressure of the ball on the bearing raceway, the stresses are pulsating type in nature, but the shape of variation with frequency ω1 was not sine-shaped (see Figure 7—left). This shape was more like an exponent-like function. Therefore, for better identification of the mean stress for secondary cycles in the direct spectral method, the exponential function was chosen for approximation.

For the analyzed case *T_B_* = *T_1_* = 0.8 s, the analytical form identified by the direct spectral method stress variation in time is given by Formula (9). The formula is defined in the range of time $t\in \left[0,T_{1}\right]$, a Concat function denotes the concatenation of two used exponential functions along time and d is a parameter of the exponential function to define the proper shape of this envelope function (for the analysed case *A_1_* = 800 MPa, *d* = 0.06).
(9)$$\sigma_{z}\left(t\right)==\left\{Concat_{t\in \left[0,\pi /\omega_{1}\right]}\left[-A_{1}exp\left(-{\left(t-\pi /\left(2\omega_{1}\right)\right)}^{2}/2d^{2}\right)\right]\;+Concat_{t\in \left[\pi /\omega_{1},2\pi /\omega_{1}\right]}\left[-A_{1}exp\left(-{\left(t-\left(3\pi \right)/(2\omega_{1})\right)}^{2}/2d^{2}\right)\right]\right\}\;+A_{2}sin\left(\omega_{2}t\right)$$

Identified cycles based on the applied spectral method are summarized in Table 3. Comparison of identified mean values and amplitudes of cycles, based on the rainflow method and the application of the Fourier method is summarized in Table 3.

The number of identified stress cycles using both methods was comparable, especially bearing in mind the fact that starting from about the 20th cycles, there were small stress amplitudes for aluminum or steel.

The values of the identified secondary stress cycles had higher values than the corresponding cycles identified by the rainflow method. The results obtained in this case using the Fourier method, combined with the direct spectral method, gave more reliable results. The value of the identified stress amplitude for the primary cycle for the rainflow method was clearly too high, bearing in mind the zero-to-compression (pulsating) nature of the course. This value obtained by the Fourier method, together with the use of the direct spectral method, was correctly identified.

Overall, the identification of cycles using the Fourier method, combined with the direct spectral method, gave better results than those obtained using the rainflow method.

## 5. Conclusions

The application of the Fourier methods for cycle identification and description for different cases of bimodal irregular stress histories was discussed in the paper. Based on this study, the following conclusions can be formulated:The direct spectral method and the modified direct spectral method, previously co-proposed by the author, together with the aspects of the use of Fourier methods discussed in this article, provides a unique alternative to the methods known in the literature for identifying and counting various types of bimodal stress variation in time (with constant or variable amplitudes; completely reversed, reversal, or pulsed type).The direct spectral method [10,11,12,13] and the modified direct spectral method [14] method are useful, especially in cases when it is known that due to the form of work of the mechanical system, the two existing components of vibrations with various frequencies can be identified, of which the one with the higher frequency has a vibration amplitude lower than a component with a lower frequency and, above all, a variable in time. In these cases, the application of the spectral methods or the rainflow method for cycle-counting seems to be unnatural.The Fourier-based identification method, applied together with the direct spectral method for bimodal waveforms, is a more natural method. For irregular waveforms with variable amplitude, it can give better results than other methods. In this case, it is worth using this method at least as a checking method in relation to other methods.Frequency analysis is often carried out using Fourier methods. The article thoroughly discusses the way the results are used and interpreted using the FFT and STFT methods. For frequency analysis of the stress with variable in time amplitude of the component, the Wavelet Transform (WT) can be applied too, as an alternative approach.When using Fourier methods in a discreet formulation, as is the case in computer analyses, the proper selection of the parameters of the analysis is crucial. For FFT analyses, they are sampling time Δt and associating it with discretization in frequency Δf. For STFT analyses, they are sampling time Δt, number of segments, number of samples per segment, time duration of the segment, and discretization in frequency Δf.

## Figures and Tables

**Figure 1 materials-16-00254-f001:**
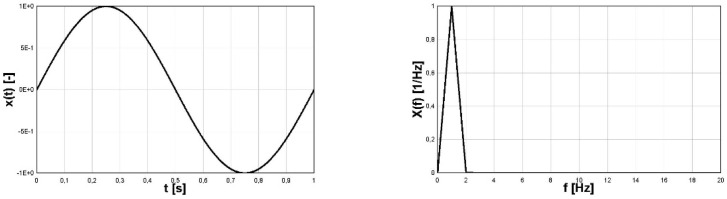
Harmonic sine function (**left**) and its FFT form (**right**).

**Figure 2 materials-16-00254-f002:**
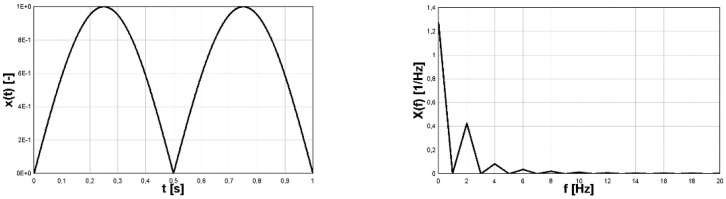
Absolute value of harmonic sine function (**left**) and its FFT form (**right**).

**Figure 3 materials-16-00254-f003:**
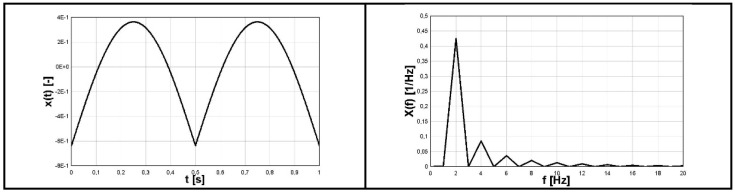
Translated absolute value of harmonic sine function (**left**) and its FFT form (**right**).

**Figure 4 materials-16-00254-f004:**
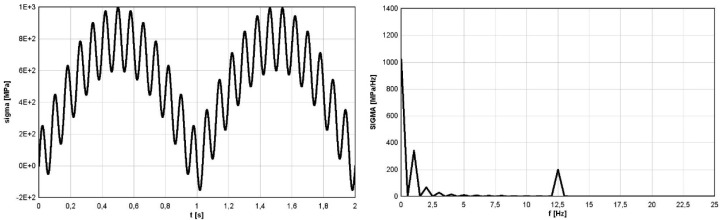
Time history of the function (8) (**left**) and its FFT form (**right**).

**Figure 5 materials-16-00254-f005:**
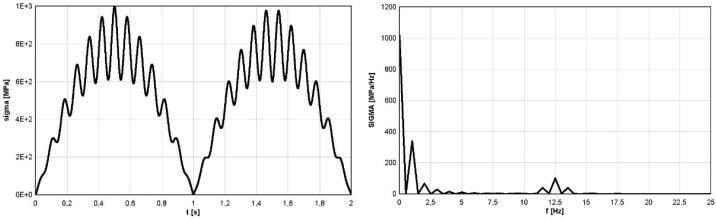
Time history of the function (9) (**left**) and its FFT form (**right**).

**Figure 6 materials-16-00254-f006:**
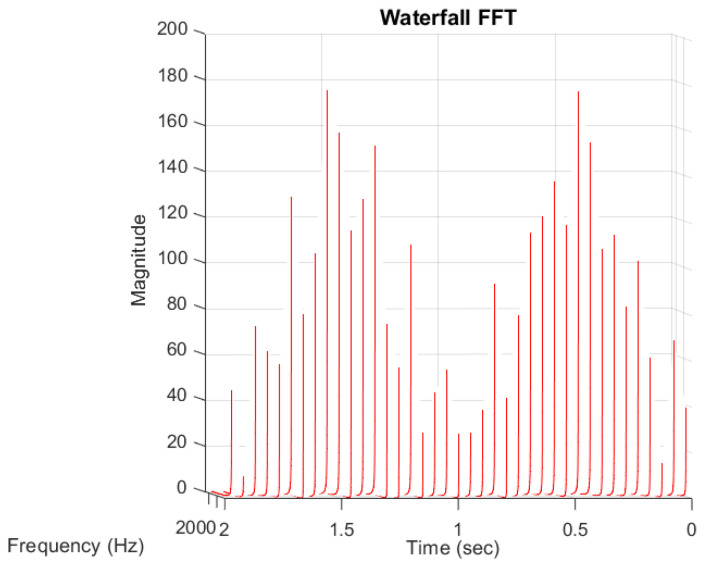
STFT form of the function (9).

**Figure 7 materials-16-00254-f007:**
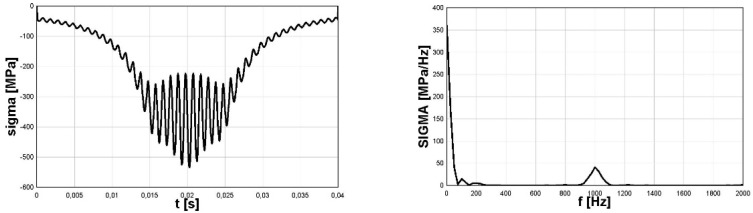
Time history of the analysezd function and its FFT form.

**Figure 8 materials-16-00254-f008:**
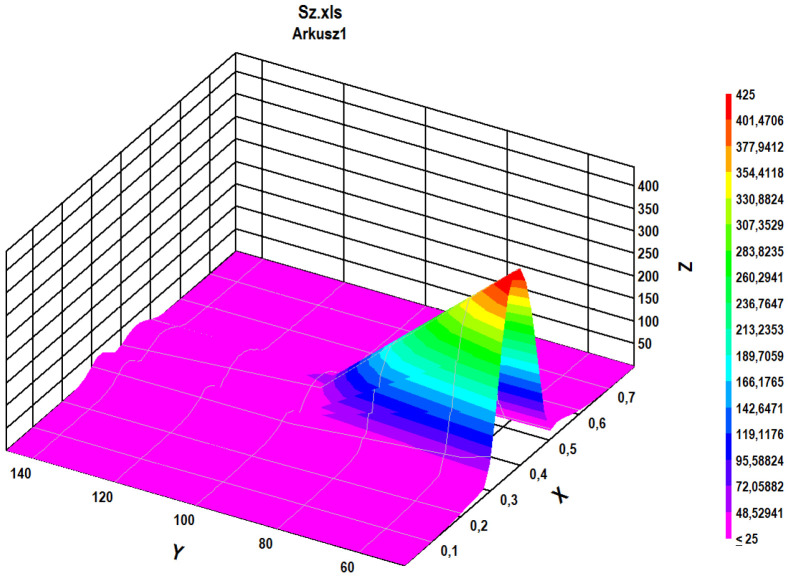
STFT form of the analyzed function shown in Figure 7 (X-time, Y-middle frequency of the STFT segment).

**Table 1 materials-16-00254-t001:** Comparison of identified mean values and amplitudes of cycles, based on rainflow and Fourier methods for stress variation defined by (8).

Cycle Type	No.	σ_a_ [MPa] FOURIER (FFT)	σ_a_ [MPa] RAINFLOW Ave.	σ_a_ [MPa] RAINFLOW Max.	σ*_m_* [MPa] FOURIER (FFT)	σ*_m_* [MPa] RAINFLOW Min.	σ*_m_* [MPa] RAINFLOW Max.
Primary cycles	1 2	501.8 501.8	562.0 562.0	576.0 576.0	501.8 501.8	421.0 421.0	474.0 474.0
Secondary cycles	0.5 1	- 200.0 199.0	300.0 184.0	300.0 197.0	- 805.2	147.0 674.0	147.0 797.0
	2	199.0	184.0	197.0	805.2	674.0	797.0
	3	199.0	184.0	197.0	792.5	674.0	797.0
	4	199.0	184.0	197.0	792.5	674.0	797.0
	5	199.0	184.0	197.0	767.3	674.0	797.0
	6	199.0	184.0	197.0	767.3	674.0	797.0
	7	199.0	184.0	197.0	730.0	674.0	797.0
	8	199.0	184.0	197.0	730.0	674.0	797.0
	9	199.0	184.0	197.0	681.2	674.0	797.0
	10	199.0	147.0	169.0	681.2	100.0	615.0
	11	199.0	158.0	169.0	621.6	100.0	615.0
	12	199.0	158.0	169.0	621.6	100.0	615.0
	13	199.0	158.0	169.0	552.3	100.0	615.0
	14	199.0	158.0	169.0	552.3	100.0	615.0
	15	199.0	158.0	169.0	474.2	100.0	615.0
	16	199.0	158.0	169.0	474.2	100.0	615.0
	17	199.0	158.0	169.0	388.7	100.0	615.0
	18	199.0	158.0	169.0	388.7	100.0	615.0
	19	199.0	158.0	169.0	297.0	100.0	615.0
	20	199.0	158.0	169.0	297.0	100.0	615.0
	21	199.0	158.0	169.0	200.6	100.0	615.0
	22	199.0	158.0	169.0	200.6	100.0	615.0
	23	199.0	158.0	169.0	101.2	100.0	615.0
	24	199.0	158.0	169.0	101.1	100.0	615.0
	25	199.0	-	-	0.0	-	-

**Table 2 materials-16-00254-t002:** Comparison of identified mean values and amplitudes of cycles based on rainflow and Fourier methods for stress variation defined by (9).

Cycle Type	No.	σ_a_ [MPa] FOURIER (STFT)	σ_a_ [MPa] RAINFLOW Ave.	σ_a_ [MPa] RAINFLOW Max.	σ*_m_* [MPa] FOURIER (STFT)	σ*_m_* [MPa] RAINFLOW Min.	σ*_m_* [MPa] RAINFLOW Max.
Primary cycles	1 2	488.15 488.15	494.0 494.0	500.0 500.0	488.15 488.15	484.0 484.0	500.0 500.0
Secondary cycles	1	178.2	174.0	189.0	785.8	777.0	789.0
	2	177.9	174.0	189.0	798.4	777.0	789.0
	3	159.7	174.0	189.0	798.4	777.0	789.0
	4	155.4	124.0	145.0	785.8	667.0	752.0
	5	154.2	124.0	145.0	675.5	667.0	752.0
	6	138.5	124.0	145.0	760.8	667.0	752.0
	7	131.7	124.0	145.0	616.4	667.0	752.0
	8	130.8	124.0	145.0	760.8	667.0	752.0
	9	123.2	124.0	145.0	675.5	667.0	752.0
	10	115.0	82.1	82.1	723.9	609.0	609.0
	11	110.6	82.1	82.1	547.6	609.0	609.0
	12	107.2	62.5	62.5	723.9	540.0	540.0
	13	103.5	62.5	62.5	470.2	540.0	540.0
	14	93.5	35.4	44.0	385.4	380.0	464.0
	15	83.5	35.4	44.0	616.4	380.0	464.0
	16	79.9	35.4	44.0	547.6	380.0	464.0
	17	75.3	35.4	44.0	294.5	380.0	464.0
	18	68.8	6.09	11.5	100.3	198.0	291.0
	19	64.3	6.09	11.5	470.2	198.0	291.0
	20	56.3	6.09	11.5	199.0	198.0	291.0
	21	47.1	6.09	11.5	100.3	198.0	291.0
	22	38.6	-	-	199.0	-	-
	23	28.5	-	-	385.4	-	-
	24	28.0	-	-	0.0	-	-
	25	15.1	-	-	294.5	-	-

**Table 3 materials-16-00254-t003:** Identified mean values and amplitudes of σ_z_ stress components.

Cycle Type	No.	σ*_z_*_,a_ [MPa] FOURIER (STFT)	σ*_z_*_,a_ [MPa] RAINFLOW Ave.	σ*_z_*_,a_ [MPa] RAINFLOW Max.	σ*_z_*_,m_ [MPa] FOURIER (STFT)	σ*_z_*_,m_ RAINFLOW Min.	σ*_z_*_,m_ RAINFLOW Max.
Primary cycle	1	420.9	654.0	654.0	−420.9	−656.0	−656.0
Secondary cycles	1	441.8	427.0	438.0	−800.0	−845.0	−815.0
	2	439.5	427.0	438.0	−756.8	−845.0	−815.0
	3	398.5	359.0	381.0	−756.8	−768.0	−705.0
	4	396.9	359.0	381.0	−640.6	−768.0	−705.0
	5	313.2	281.0	281.0	−485.2	−625.0	−625.0
	6	301.7	217.0	217.0	−640.6	−533.0	−533.0
	7	192.8	152.0	152.0	−328.9	−437.0	−437.0
	8	168.6	93.4	93.4	−485.2	−347.0	−347.0
	9	73.7	49.6	49.6	−199.5	−272.0	−272.0
	10	62.1	22.5	22.5	−328.9	−214.0	−214.0
	11	26.5	3.88	9.46	−52.6	−172.0	−37.3
	12 13 14 15 16 17 18 19 20 21 22 23 24 25 26 27 28 29 30 31 32 33 34 35 36 37 38 39 40 40.5	24.1 20.7 19.4 18.3 15.0 13.2 13.1 10.5 9.2 8.0 6.4 6.0 4.5 4.5 3.4 3.2 2.6 2.3 1.9 1.6 1.5 1.2 1.1 0.9 0.9 0.7 0.6 0.5 - -	3.88 3.88 3.88 3.88 3.88 3.88 3.88 3.88 3.88 3.88 3.88 3.88 3.88 3.88 3.88 3.88 3.88 3.88 3.88 3.88 3.88 3.88 3.88 3.88 3.88 3.88 3.88 3.88 3.88 3.88	9.46 9.46 9.46 9.46 9.46 9.46 9.46 9.46 9.46 9.46 9.46 9.46 9.46 9.46 9.46 9.46 9.46 9.46 9.46 9.46 9.46 9.46 9.46 9.46 9.46 9.46 9.46 9.46 9.46 9.46	−22.9 −108.3 −199.5 −8.9 −108.3 −52.6 −3.1 −22.9 −1.0 −8.9 −0.3 −3.1 −1.0 −0.1 −0.3 0.0 −0.1 0.0 0.0 0.0 0.0 0.0 0.0 0.0 0.0 0.0 0.0 0.0 - -	−172.0 −172.0 −172.0 −172.0 −172.0 −172.0 −172.0 −172.0 −172.0 −172.0 −172.0 −172.0 −172.0 −172.0 −172.0 −172.0 −172.0 −172.0 −172.0 −172.0 −172.0 −172.0 −172.0 −172.0 −172.0 −172.0 −172.0 −172.0 −172.0 −172.0	−1.92 −1.92 −1.92 −1.92 −1.92 −1.92 −1.92 −1.92 −1.92 −1.92 −1.92 −1.92 −1.92 −1.92 −1.92 −1.92 −1.92 −1.92 −1.92 −1.92 −1.92 −1.92 −1.92 −1.92 −1.92 −1.92 −1.92 −1.92 −1.92 −1.92

## Data Availability

The data presented in this study are available on request from the author.

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
