# Peer review of "Using the Fourier Methods for Cycle Counting of Bimodal Stress Histories with Variable in Time Amplitudes of Components"

_materials, 2022, doi:10.3390/ma16010254_

Round 1
Reviewer 1 Report
(1) Attention should be paid to the problem of regular and italic symbols. In addition, there are some grammar errors and typeset errors. Some expressions are confused, such as stress (4), signal (5), etc. Some expressions are hard, such as vertically shifted, etc.
(2) What is the subtract mean procedure? Please describe it clearly.
(3) The relative errors reaches -20.8% and 31%. It is impossible to show enough good agreement, or to be at a good level of agreement.
(4) The ring-rolling process principles, and the specific example corresponding to Fig.7-8 and Table.8, should be described in details.
(5) How to verify this work? It should be discussed in a certain.
Reviewer 2 Report
Authors insist that the paper investigates the implementation of the Fourier techniques to cycle recognition and illustration for different 10 occasions of bi-modal stress histories for the design of mechanical system such as (ball/rolling) bearing.
I checked the methodology and results. The point is a good one to use the Fourier method for analyzing the fatigue failure of ball/roller bearing due to the external repeated loads. I know external load has sinusoidal form. However, there is no clear explanation (or algorithm) why the Fourier analysis is effective for the fatigue failure of ball bearing. For example, 1). Stress ratio (maximum/minimum), 2) judgement is how (or when) to count the sinusoidal inputs that will reach to fatigue in bearing. So, it looks like a review article, not research one.
I recommend to resubmit the papers after modifying and summarizing it. My judgments come from as following:
1) First of all, structure is not good. Title need to be modified for specific purpose such as for implementation of Fourier Methods for Cycle Counting of Bi-Modal in the fatigue design of bearing. Abstract is more concise, which include your specific algorithm for judging the fatigue.
2) Introduction need more explanation (or reference) for algorithm that will judge the fatigue in bearing. It is too short to add up the more background for your research.
3) Section 3 need to add more algorithm for understanding the procedure to judge the algorithm. Results is not enough and suggest additional figures and tables for this method.
4) In section 4 title (Results and discussion of application of Fourier methods to identify stress cycles for superposition of pulsating and reversal stress components with variable amplitudes) or subtitle makes concise.
5) Conclusion also is not concise. It is required to distinctively show your research results with using some bullet. Additionally, there are a lot of spelling error. Please check
Reviewer 3 Report
In the paper, authors investigate the application of the Fourier methods to cycle identification and description for different cases of bi-modal stress histories.
The current work has lack of enough literature support. In my opinion this paper needs major revision.
Comment (1): The novelty of the investigation needs more explanation
Comment (2): Presented results should be first compared to the existing ones.
Comment (3): The results need more deep explanation
Round 2
Reviewer 2 Report
I checked the modified manuscript. All issues are raised. I recommend it as current form
Reviewer 3 Report
The authors have replied to all comments